# Access to Healthcare for Migrant Patients in Europe: Healthcare Discrimination and Translation Services

**DOI:** 10.3390/ijerph18157901

**Published:** 2021-07-26

**Authors:** Alejandro Gil-Salmerón, Konstantinos Katsas, Elena Riza, Pania Karnaki, Athena Linos

**Affiliations:** 1Polibienestar Research Institute, University of Valencia, 46010 Valencia, Spain; 2International Foundation for Integrated Care, Oxford OX2 6UD, UK; 3Institute of Preventive Medicine Environmental and Occupational Health Prolepsis, 15125 Marousi, Greece; k.katsas@prolepsis.gr (K.K.); p.karnaki@prolepsis.gr (P.K.); a.linos@prolepsis.gr (A.L.); 4Department of Hygiene Epidemiology, Medical Statistics Medical School National, Kapodistrian University of Athens, 11527 Athens, Greece; eriza@med.uoa.gr

**Keywords:** migrant patients, healthcare access, discrimination, translation services

## Abstract

Background: Discrimination based on ethnicity and the lack of translation services in healthcare have been identified as main barriers to healthcare access. However, the actual experiences of migrant patients in Europe are rarely present in the literature. Objectives: The aim of this study was to assess healthcare discrimination as perceived by migrants themselves and the availability of translation services in the healthcare systems of Europe. Methods: A total of 1407 migrants in 10 European Union countries (consortium members of the Mig-HealthCare project) were surveyed concerning healthcare discrimination, access to healthcare services, and need of translation services using an interviewer-administered questionnaire. Migrants in three countries were excluded from the analysis, due to small sample size, and the new sample consisted of N = 1294 migrants. Descriptive statistics and multivariable regression analyses were conducted to investigate the risk factors on perceived healthcare discrimination for migrants and refugees in the EU. Results: Mean age was 32 (±11) years and 816 (63.26%) participants were males. The majority came from Syria, Afghanistan, Iraq, Nigeria, and Iran. Older migrants reported better treatment experience. Migrants in Italy (0.191; 95% CI [0.029, 0.352]) and Austria (0.167; 95% CI [0.012, 0.323]) scored higher in the Discrimination Scale to Medical Settings (DMS) compared with Spain. Additionally, migrants with better mental health scored lower in the DMS scale (0.994; 95% CI [0.993, 0.996]), while those with no legal permission in Greece tended to perceive more healthcare discrimination compared with migrants with some kind of permission (1.384; 95% CI [1.189, 1.611]), as opposed to Austria (0.763; 95% CI [0.632, 0.922]). Female migrants had higher odds of needing healthcare assistance but not being able to access them compared with males (1.613; 95% CI [1.183, 2.199]). Finally, migrants with chronic problems had the highest odds of needing and not having access to healthcare services compared with migrants who had other health problems (3.292; 95% CI [1.585, 6.837]). Conclusions: Development of culturally sensitive and linguistically diverse healthcare services should be one of the main aims of relevant health policies and strategies at the European level in order to respond to the unmet needs of the migrant population.

## 1. Introduction

Given the continuous immigration flows to Europe and the increased numbers of migrants arriving to EU member states [1], issues related to the more equitable and effective delivery of healthcare become a high priority for EU member states. Increasing access to basic health services for the migrant population in host countries is of utmost importance. The scant evidence that exists on healthcare access for migrants in Europe makes it difficult to make comparisons between systems and countries or to support public policy decision making [2]. Although the lack of language services overall and the fear of being ethnically discriminated within the healthcare setting are highlighted in the literature as prevalent problems for migrants when accessing healthcare [3], less is known about the experiences of the migrant population once they actually access the healthcare services in Europe. 

The rights of refugees, asylum seekers, and migrants to healthcare access vary across European countries in terms of regulation and laws [4]. Even when access to healthcare is granted by law, research suggests that migrant groups, in particular asylum seekers [5] and undocumented migrants [6,7], face several obstacles when trying to access healthcare. In this regard, the difficulty to communicate effectively in a host country’s language can impede access to healthcare services [8]. The provision of translation services can help to overcome these communication barriers; however, the availability of free and accessible interpretation services is highly variable across Europe [9]. Studies indicate that the presence of professional interpreters can improve both the quality of care [10] as well as patient satisfaction. Alternatively, inefficient communication between healthcare providers and patients increases the risk of misunderstanding and misdiagnosis [11]. Moreover, patients who face communication difficulties visit their healthcare provider less often [12] and are less compliant with medication and treatment advice [13]. The lack of professional interpreters has been associated with unnecessary, extensive, and potentially harmful medical exams, treatments, and hospitalisations [14,15]. 

Furthermore, a large-scale study (QUALICOPC) conducted in 31 European countries showed that most differences on perceived healthcare discrimination were found between the native inhabitants of a country and first-generation migrants, reporting more discrimination within healthcare settings for the migrant population [16]. In this regard, previous studies have identified that experiencing ethnic discrimination in the healthcare setting is associated with poor quality of care [17], highest levels of distrust in healthcare providers as well as distrust in the healthcare system overall [18,19]. Moreover, different studies have also found an association between experiencing healthcare discrimination and decreased healthcare utilization and participation in preventive services. Studies have also indicated that discrimination within healthcare settings is related to increased emergency department visits and hospital admissions [20]. Moreover, experiencing healthcare discrimination has a detrimental impact on health outcomes. As a result of racial discrimination, studies have shown that migrant patients with diabetes do not receive appropriate diabetes care [21] and experience higher risk of diabetes comorbidities [22]. Moreover, racial discrimination in healthcare is also associated with lower medication adherence in ethnic diverse patients taking hypertension medication [23,24].

### Mig-HealthCare Background

The Mig-HealthCare project was a three-year project launched in May 2017 [25]. The project’s main scope was to provide evidence-based information and practical guidance to primary healthcare professionals, primarily in the EU member states, on how to best address the health issues of refugee and migrant populations. 

In order to support public policy decision making to reduce obstacles in healthcare access, and thus reduce health disparities for migrant populations in Europe, it is important to study the migrant patient experience as a result of them accessing healthcare services. Given the worse health outcomes, healthcare misuse and receiving lower quality of care associated with the perception of ethnic discrimination in healthcare and the lack of interpretation services, generating a greater understanding of the patient experience of the migrant groups in Europe is important to plan future tailored healthcare systems and services [26]. This paper, has collected and analysed migrant patient experiences in 10 European countries (Austria, Bulgaria, Cyprus, France, Germany, Greece, Italy, Malta, Spain, and Sweden). The primary aim of this study is to assess the perception of ethnic discrimination within healthcare services in Europe while having a comparative framework to understand the commonalities and differences across European countries by country of origin and language proficiency of migrants. The secondary aim is to assess the need for translation services when accessing healthcare services and identify factors that predict the feeling of being ethnically discriminated when accessing healthcare in Europe. 

## 2. Methodology

### Design and Sample

This study was a cross sectional survey that included 1407 participants. A detailed description of the methodology is reported elsewhere [27]. Participants were recruited via the snowball sampling method in 10 European countries (Austria, Bulgaria, Cyprus, France, Germany, Greece, Italy, Malta, Spain, and Sweden). Participants were eligible if they were aged 18 years old or over, had resided in the country of interview for a period of 6 months to 5 years, and were able to both understand the study goals and give consent for participation in the survey. In the case where study participants could not communicate in the language of the host country of interview, the assistance of an accredited interpreter was sought and provided to complete the questionnaire. Demographics of the total sample (N = 1407) are described in Appendix A. We excluded migrants from France, Germany, and Malta from our analysis, due to a small sample size in these countries and the new sample consisted of N = 1294 migrants.

Participation in the study was purely voluntary, with the sample comprising refugees/migrants who were visiting health and social care delivery services in which Mig-HealthCare partners operated (such as health centres in refugee camps, primary healthcare centres, community centres, and NGO clinics). The specifically designed questionnaire was tested and translated into Arabic, Farsi, Dari, Pashto, Somali as well as into the languages of the partner countries (migrant and refugee host countries). All interviewers were trained prior to the initiation of the study. Data collection took place from April 2018 to September 2019, after being approved by the Ethical Committee of both the University of Valencia and the National and Kapodistrian University of Athens, Medical School. Additional ethical approvals were obtained as necessary by each participating partner organization. No identifiable personal data were collected, and a unique identity was assigned to each study participant which was only available to the main researchers. The purpose of the study and the data collection method was clearly explained to each study participant and informed consent was sought and received before participation.

## 3. Measures

### 3.1. Discrimination in Medical Settings (DMS)

Ethnic discrimination experienced in medical settings was measured through the Discrimination Scale in Medical Settings (DMS scale) [28]. Participants were asked the following questions: “When getting healthcare of any kind, have you ever had any of the following things happen to you because of your race or ethnicity?”, with seven adapted items: (1) you are treated with less courtesy than other people, (2) you are treated with less respect than other people, (3) you receive poorer service than others, (4) a doctor or nurse acts as if they think you are not smart, (5) a doctor or nurse acts as if they are afraid of you, (6) a doctor or nurse acts as if they are better than you, and (7) you feel like a doctor or nurse is not listening to what you were saying. Response categories were 1 = never, 2 = rarely, 3 = sometimes, 4 = most of the time, and 5 = always. Then, all seven questions were summed and a mean score on the entire scale was computed, with higher scores indicating more perceived discrimination (range from 1–5 units). The DMS scale was evaluated for reliability. Pearson correlations between DMS score and its seven component items were positive and larger than 0.3 and the diagonal Cronbach’s α results scored excellent (more than 0.9), therefore confirming the reliability of the DMS scale (Appendix A). A mean score on the entire scale was computed, with higher scores indicating more perceived discrimination.

### 3.2. Mental and Physical Health Status 

For this study, the relevant to mental health scale of the Short Form 36 (SF-36) [29] was used to provide an assessment of mental health status. The following five questions were used to assess psychological distress and well-being, scoring from 0 (low) to 100 (high): “Have you been a very nervous person? Have you felt so down in the dumps that nothing could cheer you up? Have you felt calm and peaceful? Have you felt downhearted and blue? Have you been a happy person?”. The total mental health score was evaluated for reliability. Pearson correlations between the mental health score and its five component items were positive and larger than 0.3 and total Cronbach’s α was acceptable (>0.7), therefore confirming the reliability of the mental health score (Appendix A).

To assess physical health status, participants were asked about chronic conditions. A list of 22 choices was provided (multiple options possible) following the question: “Do you suffer from any of the following chronic diseases or long-term conditions? (tick all that apply)”.

### 3.3. Accessibility to Healthcare Services and Translation Services 

Accessibility to healthcare services was measured with the following question “Need to use healthcare services the last 6 months”, respondents were given the following response options: “Needed and did not have access”, “Needed and had access” and “Did not need”. Two more indicators of accessibility to healthcare were also asked. Participants were asked if they perceived having worse access to healthcare services compared with local people with the response categories being “yes” and “no”. Moreover, participants were asked about the need of translation services during their medical visits with the response scale being never, few times, most times, and always. 

### 3.4. Sociodemographic Characteristics 

The following variables were included to further elaborate on the study sample characteristics: age, sex, country of origin, country of interview, number of years in the education system, having children, speaking language of the country of interview, and legal situation in the country.

### 3.5. Statistical Analysis

A descriptive analysis was performed for all available data. We conducted linear regression analysis to investigate variations in DMS scale (dependent variable) by country of interview, country of origin, and socioeconomics. Confounding factors were examined by applying three different linear models, starting from the univariable model (forward procedure). To investigate relevant changes in DMS score by legal situation in the country and health status, we selected a negative binomial model due to poor fit of the linear (Poisson distribution of the residuals) and Poisson (overdispersion) models. Finally, a multivariable logistic model was used to compare the odds of having access between migrants with different health status, country of interview, and kind of permission to stay in the country. All variables in each model were initially conceptualized by the perspective of clinical interest. Following this, the final models with the best fit were selected using Collet’s method, based on Akaike information criterion (AIC). All variables in the models were tested for collinearity. The level of statistical significance was defined as alpha = 0.05.

## 4. Results

The general demographics of the study participants are presented in Table 1. In Table 2 we observed a high DMS score in Greece, Italy, Cyprus, and Austria. The lowest score was reported in Spain. Migrants from Afghanistan tended to score higher in the DMS scale.

In Table 3, perceived discrimination in migrant women is presented by country of interview, country of origin, and age. We observed a high DMS score in migrant women in Cyprus and Greece. The lowest score was reported in Spain (same as Table 2). Migrant women from Iran scored significantly higher in the DMS scale compared with other countries of origin. Higher age indicated lower DMS score among migrant women.

Migrants in Greece reported needing and not having access to healthcare services less frequently, compared with other countries (Table 4). Almost all migrants in Bulgaria and the majority in Italy and Greece required a translator, while the lowest percentage of requiring a translator was observed in Spain.

Linear regression models are presented in Table 5 to observe associations in DMS scale by country of interview, country of origin, and socioeconomic status. In Model 3, the natural logarithmic transformation of the DMS score as the dependent variable was used due to poor fit of the standard linear regression model. Migrants in Italy and Austria scored higher in the DMS scale compared with Spain (Models 1–3). The DMS score was lower for older participants and for those with more years of education (Models 2,3). Migrants from Nigeria, Syria, Iraq, and other countries scored lower in the DMS scale compared with migrants from Iran. Age was significantly associated with DMS scores, with younger migrants scoring higher DMS (−0.006; 95% CI [−0.012, −0.001]).

Speaking the language of the country of interview was negatively associated with the DMS score in the univariate model (−0.093; 95% CI [−0.164, −0.021]) (not shown in the Table) and it was not statistically significant in the multivariate model. This could be explained perhaps due to the potential confounding effect of country of interview and country of origin (Appendix A).

DMS was transformed into an integer scale (e.g., from 1.257 to 1257) due to poor fit in the linear model and Poisson distribution of the residuals. We used negative binomial regression due to increased dispersion in the Poisson model (Table 6). Older migrants reported better treatment experience (Models 1,2). Migrants with better self-perceived mental health scored lower in the DMS scale (0.994; 95% CI [0.993, 0.996]).

In Model 2, we added an interaction term for country of interview to include other kinds of legal permission. Migrants with no other kind of residence permission in Greece had higher DMS scores compared with migrants with some kind of permission (1.384; 95% CI [1.189, 1.611]). Migrants with no kind of permission in Austria had lower DMS score by 24% compared with migrants who had some kind of permission in Austria (0.763; 95% CI [0.632, 0.922]).

Logistic regression analysis was performed to investigate the likelihood of having access to healthcare services (Table 7). Migrants in Greece were more likely to need and not have access to healthcare services compared with Spain (0.293; 95% CI [0.166, 0.516]). Female migrants had 60% higher odds of needing and not having access to healthcare services compared with males (1.613; 95% CI [1.183, 2.199]). Migrants with health problems (chronic problems from injury/accidents, gastrointestinal disease, diabetes, skin disease, headaches/migraines, and diseases related to bone and muscle) were more likely to needing and not having had access to healthcare services compared with healthy migrants. Migrants with chronic problems had the highest odds of needing and not having access to healthcare services compared with other health problems (3.292; 95% CI [1.585, 6.837]). Migrants with gastrointestinal disease or diabetes had higher odds ratios compared with migrants with skin diseases, headaches, or migraines, and diseases related to bone and muscle.

## 5. Discussion

To our knowledge, this is one of the first studies to assess the patient experience of the migrant population across 10 countries in Europe. The main finding from this study is that better self-reported mental health outcomes as measured by the SF-36 relevant mental health scale were associated with lower perceived discrimination in medical settings. Moreover, migrant women were more likely to not be able to access healthcare services when needed. Likewise, the same findings were reported for migrants suffering from chronic illnesses. Finally, older migrants reported higher feelings of health discrimination.

To our knowledge, this is the first study using the DMS questionnaire among a multi-cultural group of migrants in Europe. Two previous studies validated the DMS in two ethnic minorities samples in the United States [21,30]. Similar to these studies, we evaluated the reliability of the scale by calculating Cronbach’s α and Pearson correlations between the DMS score and its seven component items (r > 0.3). Although there is a large number of migrants arriving to Europe, no studies have been conducted assessing ethnic discrimination in healthcare services. For this reason, future studies should explore psychometric properties of this scale among specific migrant groups, different healthcare settings, and across Europe. In this regard, the results of this study contribute to healthcare systems efforts to assess and address healthcare discrimination in the migrant population [31].

Moreover, findings indicate that the migrant population reported higher levels of healthcare discrimination in Greece, Italy, Cyprus, and Austria and lower levels in Spain (*p* < 0.001). Additionally, migrants from Afghanistan tended to score higher in the DMS scale (*p* < 0.05). The lower level of healthcare discrimination reported in Spain may be explained by some characteristics of the study population because 186 (92.08%) participants were from South America and were Spanish speakers. Therefore, our results demonstrate a high relationship between the need for translators and the feelings of being discriminated when accessing healthcare services, making the role of translation services highly relevant for the quality of care for migrant populations. Recent evidence has also suggested additional benefits at system and professional levels, such as cost-saving for the healthcare system, and reducing difficulties during the appointments [8].

The results of our study also show significant differences of perceived ethnic discrimination in the healthcare system especially regarding age. Here, younger migrants reported worse treatment experience in our sample from 7 EU countries. A recent systematic literature review by Robards and colleagues [32] found that for marginalized young people, the decision to access health services is affected by previous bad experiences during which they felt treated differently and with disrespect by healthcare professionals. In the same article, the study population highlighted different actions to be considered in the delivery of healthcare for migrant young groups such as culturally appropriate services, cultural sensitivity of staff, and the use of interpreters.

Our study also highlighted the vulnerability of migrant women regarding both the lack of access to healthcare services and the perception of higher discrimination within healthcare services. In addition to the common barriers of migrants in accessing healthcare [33], the results of this study could be explained by differences in care seeking behaviour of this group [34] or in health services-related factors [35]. For this reason, health interventions aiming to mitigate gender-driven inequalities in accessing quality healthcare have to be put in practice in the European context [36].

The management of chronic diseases in European migrants and refugees has been identified as a priority for health service provision [27]. However, the results of this study show that migrant chronic patients reported the highest odds for needing but not having access to healthcare services. This could be explained by the higher contact frequency that chronic patients have with the healthcare systems, which may increase their health literacy and empowerment, making them more aware of their needs, their rights, and their expectations from the system. Hence, this group could be more perceptive to discrimination during their medical visits. Moreover, more frequent health visits could increase the likelihood to be exposed to experiences of healthcare discrimination towards them. Another reason may be that frequent visits expose general inefficiencies of the healthcare system and provision (such as lack of personnel or equipment) that are more attenuated in these groups.

Additionally, in our study differences were found concerning healthcare discrimination towards undocumented migrants depending on the host country. In this regard, undocumented migrants in Greece (1.384; 95% CI [1.189, 1.611]) had higher DMS scores compared with migrants with legal status. However, the opposite trend was reported in Austria, where undocumented migrants had a lower DMS score by 24% compared with migrants with legal status (0.763; 95% CI [0.632, 0.922]). In this regard, Robertshaw and colleagues [37] found that immigration status and legislative policy are a challenge for the provision of healthcare by creating or reinforcing vulnerability of marginalized groups [38]. However, the results of our study could be interpreted in the light of the results of Dauvrin and colleagues [39] who also reported insufficiencies in the actual delivery of care for undocumented migrants despite the variations in healthcare entitlement related to the immigration status across Europe, suggesting that even in countries with “minimum rights”, health professionals may consider treating undocumented migrants more important than abiding by law (“pragmatic health professional”). For that reason, our results might be outlining a complex interplay of different factors that might be worsening the provision of healthcare for migrant patients in the host country beyond their legal status, as this is not a general result and differences appear between countries in our study.

Although specific migrant groups have reported experiencing discrimination in healthcare, ethnic discrimination, and translation services, these are still under-researched topics in Europe. In this regard, our results could be also interpreted from a structural and organizational point of view for healthcare delivery. Indeed, King and colleagues [40] argue how (1) compulsory assigned residency, (2) resources (including language skills), and (3) freedom of movement (related to documented/undocumented status) could be consolidating heavy and stable forms of devaluation, reification, and stigma, denying access to healthcare for certain groups with negative consequences on the health of migrants.

This study shows the experiences of migrants and refugees between 2018 and 2019. The inequalities and vulnerabilities shown in this study may have been further amplified by the COVID-19 pandemic, which was not captured in our study but should be briefly discussed. Even though healthcare services face an unprecedented demand, groups such as undocumented migrants and migrant women are still facing barriers to access appropriate quality care that contributes to poorer health outcomes [41,42]. Moreover, epidemiological data shows that COVID-19 disproportionately affects patients who have chronic conditions and underlying comorbidities [43]. This is in line with the report of Fiorini and colleagues [38], indicating that due to limited access to appropriate care and quality of care, migrant and refugee populations, and specifically some groups such as undocumented female or chronic patients, might experience higher morbidity and mortality during the COVID-19 pandemic,

To the best of our knowledge, this is the first study assessing the migrant patient experience with a multicultural sampling in seven EU Member states regarding healthcare discrimination and the need for translation services. Another strength includes the range of data and populations investigated allowing for comparative and intersectional analyses across many dimensions of discrimination in healthcare services. On the other hand, limitations include relying on self-reported information which may introduce reporting bias. Furthermore, study participants varied greatly in country of origin, duration of stay in country of interview, and integration phase. The use of interpreters may have introduced additional information bias, and cultural barriers in female representation in the survey for some countries (such as Afghanistan) may have biased their responses. Moreover, considering the convenience sampling method used, respondents may have given socially desirable answers, leading to underestimation of healthcare discrimination. Finally, as this is a cross-sectional study which relies on a non-random sample, causal relationships cannot be established.

## 6. Conclusions

In conclusion, migrant and refugee patients reported unequal access to healthcare and perceived discrimination when they did access services. We found that country of origin and not speaking the host country language were associated with increased discrimination in healthcare provision. Language communication support and cultural mediation in healthcare services will facilitate healthcare access. Moreover, younger migrants, migrant women, and chronic migrant patients reported experiencing higher discrimination within the healthcare services. The need for health interventions to address the unequal access of these groups to appropriate care and quality of care is now further stressed in the light of the COVID-19 pandemic that has amplified shortcomings in the provision of care to these groups. Finally, it is important to conduct cohort studies monitoring healthcare access and perceived discrimination towards different migrant groups as part of the quality control of healthcare provision.

## Figures and Tables

**Table 1 ijerph-18-07901-t001:** Characteristics of migrants and refugees, except those from Germany, France, and Malta (N = 1294).

	N (%)
Country of interview	
Austria	126 (9.74)
Bulgaria	226 (17.47)
Cyprus	110 (8.50)
Greece	255 (19.71)
Italy	271 (20.94)
Spain	202 (15.61)
Sweden	104 (8.04)
Gender (males)	816 (63.26)
Country of origin	
Afghanistan	187 (14.55)
Iran	48 (3.74)
Iraq	122 (9.49)
Nigeria	115 (8.95)
Syria	281 (21.87)
Other	532 (41.40)
Having at least one child	535 (50.71)
Asylum (yes)	320 (27.80)
Other kind of permission (yes)	562 (45.03)
Speaking country of interview language (yes)	428 (38.73)
Comorbidity	
Not having a disease or chronic condition	653 (50.46)
Having one disease or chronic condition	311 (24.03)
Having at least two diseases or chronic conditions	330 (25.5)
	Mean ± standard deviation
Age (years)	32 ± 11
Education (years)	8.9 ± 5.1

**Table 2 ijerph-18-07901-t002:** Discrimination Scale to Medical Settings by country of interview, country of origin, and communication skills. Migrants and refugees from Germany, France, and Malta were excluded from the analysis.

	N	Median	Interquartile Range	
Country of interview ***				
Austria	91	1.286	1	2.286
Bulgaria	93	1	1	2
Cyprus	63	1.286	1	2.286
Greece	178	1.286	1	2.571
Italy	115	1.286	1	2.286
Spain	160	1	1	1.286
Sweden	48	1.071	1	1.786
Country of origin **				
Afghanistan	110	1.571	1	3
Iran	29	2	1	3
Iraq	72	1.071	1	2.071
Nigeria	54	1.143	1	2
Syria	169	1	1	2
Other	309	1	1	1.857
Ability to communicate in host country language *				
Not speaking country of interview language	104	1	1	2.143
Speaking country of interview language	644	1	1	1.714

*** *p* < 0.001, ** *p* < 0.05, * *p* < 0.1.

**Table 3 ijerph-18-07901-t003:** Discrimination Scale to Medical Settings by country of interview, country of origin, and age in migrant women. Migrants and refugees from Germany, France, and Malta were excluded from the analysis.

	N	Median	Interquartile Range	
Country of interview				
Austria	45	1.286	1	2
Bulgaria	47	1	1	2
Cyprus	31	1.714	1	2.429
Greece	101	1.571	1	2.857
Italy	94	1.286	1	2
Spain	87	1	1	1.286
Sweden	20	1.357	1	2.5
Country of origin				
Afghanistan	75	1.571	1	3
Iran	19	2.714	1	3.143
Iraq	27	1.286	1	2.429
Nigeria	30	1	1	1.286
Syria	76	1.071	1	2.429
Other	196	1	1	1.929
		Correlation coefficient		
Age (years)	301	−0.1937		

All *p* < 0.001.

**Table 4 ijerph-18-07901-t004:** Access to healthcare services and requirement of a translator by country of interview. Migrants and refugees from Germany, France, and Malta were excluded from the analysis.

	Austria	Bulgaria	Cyprus	Greece	Italy	Spain	Sweden
Need to use healthcare services the last 6 months (%)							
Needed and did not have access	59.20	54.26	41.41	27.83	48.24	45.77	39.42
Needed and had access	11.20	0.0	19.19	36.52	1.96	27.36	19.23
Did not need	29.60	45.74	39.39	35.65	49.8	26.87	41.35
Worse access to healthcare services compared with local people (% yes)	26.09	32.57	25.56	42.52	15.05	19.9	20.19
Need for a translator							
Never	33.05	0.91	32.63	12.95	9.47	74.73	32.67
Few times	38.14	1.82	18.95	11.16	32.58	16.67	15.84
Most times	16.10	25.91	24.21	22.32	24.62	2.69	18.81
Always	12.71	71.36	24.21	53.57	33.33	5.91	32.67

All *p* < 0.001.

**Table 5 ijerph-18-07901-t005:** Multivariable linear regression models investigating the impact of country of interview, country of origin, and socioeconomic characteristics on Discrimination Scale to Medical Settings (dependent variable) in migrants and refugees. Migrants and refugees from Germany, France, and Malta were excluded from the analysis.

	Model 1 (N = 748)			Model 2 (N = 569)			Model 3 (N = 497)		
	Estimate	95% CI		Estimate	95% CI		Estimate	95% CI	
Austria as country of interview	0.318 **	0.080	0.556	0.239 *	−0.034	0.511	0.167 **	0.012	0.323
Bulgaria as country of interview	0.261 *	0.025	0.497	0.092	−0.222	0.406	0.031	−0.173	0.234
Cyprus as country of interview	0.384 **	0.115	0.654	0.185	−0.156	0.527	0.110	−0.090	0.309
Greece as country of interview	0.588 ***	0.391	0.786	0.276 *	−0.030	0.581	0.140	−0.047	0.328
Italy as country of interview	0.369 **	0.147	0.590	0.347 **	0.044	0.651	0.224 **	0.062	0.386
Sweden as country of interview	0.183	−0.116	0.481	0.211	−0.120	0.542	0.136	−0.040	0.311
Age (years)				−0.006 **	−0.012	−0.001	−0.003 *	−0.007	−0.0004
Females				0.009	−0.138	0.156	−0.013	−0.094	0.068
Education (years)				−0.019 **	−0.034	−0.003	−0.009 **	−0.018	−0.001
Not having children				−0.002	−0.146	0.151	−0.017	−0.096	0.063
Afghanistan as country of origin				−0.176	−0.566	0.214	−0.112	−0.384	0.160
Iraq as country of origin				−0.418 **	−0.836	−0.0004	−0.158	−0.404	0.089
Nigeria as country of origin				−0.606 **	−1.100	−0.111	−0.276 *	−0.554	0.003
Syria as country of origin				−0.608 **	−0.987	−0.229	−0.253 **	−0.479	−0.027
Other as country of origin				−0.427 **	−0.830	−0.024	−0.170	−0.408	0.068
Speaking country of interview language							−0.006	−0.103	0.115

Compared with Spain (reference group for country of interview) and with Iran (reference group for country of origin). *** *p* < 0.001, ** *p* < 0.05, * *p* < 0.1. The natural logarithmic transformation for DMS score (dependent variable) was used in Model 3 due to poor fit of the standard linear regression model.

**Table 6 ijerph-18-07901-t006:** Negative binomial regression investigating the impact of country of interview, permission to stay, and health status on Discrimination Scale to Medical Settings (dependent variable) in migrants and refugees. Estimates are presented as incidence rate ratio. Migrants and refugees from Germany, France, and Malta were excluded from the analysis.

	Model 1 (N = 614)			Model 2 (N = 614)		
	IRR	95% CI		IRR	95% CI	
Not having asylum	1.052	0.971	1.139	1.041	0.957	1.132
Mental Health Score	0.994 ***	0.992	0.995	0.994 ***	0.993	0.996
Age (years)	0.995 **	0.992	0.998	0.996 **	0.993	0.999
Females	0.957	0.889	1.030	0.978	0.910	1.052
Having one disease or chronic condition (morbidity)	1.000	0.889	1.030	0.981	0.893	1.078
Having at least two diseases or chronic conditions (comorbidity)	1.093 *	0.998	1.196	1.041	0.949	1.142
Not having other kind of permission in Austria				0.763 **	0.632	0.922
Not having other kind of permission in Bulgaria				1.019	0.886	1.173
Not having other kind of permission in Cyprus				1.165	0.906	1.498
Not having other kind of permission in Greece				1.384 ***	1.189	1.611
Not having other kind of permission in Italy				1.247 *	0.993	1.173
Not having other kind of permission in Spain				1.019	0.886	1.173
Not having other kind of permission in Sweden				0.882	0.683	1.138
Not having other kind of permission in total	1.047	0.974	1.126			

*** *p* < 0.001, ** *p* < 0.05, * *p* < 0.1. Model 1 is adjusted for country of interview. In Model 2, interaction terms for having another kind of asylum with country of interview were inserted. Not having another kind of permission in Bulgaria was omitted because all migrants in Bulgaria did not have other kinds of permission.

**Table 7 ijerph-18-07901-t007:** Multivariable logistic regression investigating the impact of country of interview, need of a translator, permission to stay, and health status on access to healthcare services (dependent variable) in migrants and refugees. Migrants and refugees from Germany, France, and Malta were excluded from the analysis.

N = 899	Odds Ratio	95% CI	
Austria as country of interview	1.378	0.804	2.362
Bulgaria as country of interview	1.301	0.707	2.393
Cyprus as country of interview	0.537 *	0.278	1.039
Greece as country of interview	0.293 ***	0.166	0.516
Italy as country of interview	0.976	0.576	1.653
Sweden as country of interview	0.597 *	0.340	1.047
Age (years)	0.998	0.984	1.013
Females	1.613 **	1.183	2.199
Need for a translator (yes)	1.209	0.826	1.770
Not having asylum	0.798	0.573	1.112
Not having other kind of permission	0.820	0.592	1.137
Mental Health Score	1.000	0.993	1.007
Chronic problems from injury/accidents	3.292 ***	1.585	6.837
Gastrointestinal disease	2.917 ***	1.556	5.468
Diabetes	2.912 **	1.370	6.190
Skin disease	1.912 **	1.052	3.475
Disease related to bone and muscle	1.907 **	1.030	3.531
Headaches/migraines	1.643 **	1.084	2.489

Compared with migrants in Spain as reference group for country of interview *** *p* < 0.001, ** *p* < 0.05, * *p* < 0.1.

## Data Availability

Limited dataset used for this analysis is available upon reasonable request from the Mig-HealthCare consortium.

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
