# Peer review of "Access to Healthcare for Migrant Patients in Europe: Healthcare Discrimination and Translation Services"

_ijerph, 2021, doi:10.3390/ijerph18157901_

Round 1

Reviewer 1 Report

The study represents an original contribution to the understanding of migrant population' s perceived discrimination in the health sector. The results are clearly presented and the methods Clearly explained. The article is well written and the structure clear.

Author Response

Thank you very much for your review and support. 

Reviewer 2 Report

I wish to express my sincere appreciation to you for this original scientific contribution. Great job. Congratulations!

Author Response

Thank you very much. We really appreciate your nice words. 

Reviewer 3 Report

This is a great paper. Full of potential. Useful. I want to see it published. But empirics is not robust. Great research question, but a weak sample. Making results not reliable. 

It is well written, but with poor reference to the literature. 

The paper deserves to be published. It may offer a serious contribution to the existing literature. But its empirical findings need to be much stronger. Simple suggestions: cut Germany, cut Malta and cut France. You can not treat together Germany and Austria. You can not make an analysis on a so small sample of migrants in Germany, Malta and France. 

Please add as an appendix the full questionnaire (this is a requirement)

Please add as an appendix the demographics (age, gender, country of origin) of the sample (table 1 is not enough)

Please add in 3.1 a full presentation of the DMS scale and its questions (“eg” is not enough to present your main dependent variable), your main dependent variable. Add the table with Cronbach diagonal results, question by question. Include only cases for countries where you have a large enough sample (so to say exclude Austria, Malta and France)

Please add in 3.1 the table with Cronbach diagonal results, question by question of the SF36 scale. Include only cases for countries where you have a large enough sample (so to say exclude Germany, Malta and France)

In 3.5 it is not necessary to say you have used Stata. 

Concerning the main results, please show your results (descriptives) on the migrant women perceived discrimination by Country and by age

Line 347: not useful here to refer to psychometric properties. It is confusing. 

Concerning undocumented migrants results are robust (but exclude France, Germany & Malta). But interpretation is poor. I would look at strong interpretation in migration studies: not to waste your time, I suggest you to check the recent contribution from Desmond King et al. pointing at explaining discrimination of undocumented migrants in health (as well as school and other public welfare services in European member states in terms of institutional differentialism (or institutional differential treatment) concerning (1) compulsory assigned residency, (2) resources (included language skills) and (3) freedom of movements (related to documented/undocumented status). These factors are keys to create a socialization process towards institutional skills. It is not necessary to follow the whole theory, but just to point on the fact that serious differential treatment of these 3 spheres consolidates heavy and stable forms of devaluation, reification and stigma (King, D. et al., 2017, Assimilation, Security, and Borders in the Member States. In King, D. and P. L. Galès, Reconfiguring States in Crisis. Oxford University Press), pp. 428–50.

Line 395 I will not use the concept of readiness to accommodate. It seems tautological to me. Why health system they do not accommodate undocumented migrants? Because they are not ready to. Poor explanation. There are three main ways to look at health system capacity to cope with undocumented migrants in public health literature: the first is related to the organizational dimension – it has proportionally received the largest amount of attention by the literature explaining the strategic decision-making behaviour among NGOs, Unions, Churches and pro-migrant interest groups that seek to influence system openness and response to undocumented migrants health demand. The second set of factors relates to the path-dependent universalism of the health system and to the rules and boundary mechanisms that have contributed to the construction of its reception rules via opposition/distinction of deserving “clients”, see Vanhala (2009). The third set of factors has been explored by authors looking for resources for adaptation and inclusionary improvement within this system (Jacquot in the Journal of European Public Policy 2014 has discussed this with reference to your same main discriminated groups: women and minorities) 

line 418: attention not to blame the victims

line 456: ‘convenience sampling’ is not clear, it requires to be defined.

line 457 the relation between sampling and socially desirable answers is not clear; usually, the link is with the interview method, not with the sampling technique

Please clearly state if your data are publicly available, and where they can be found and requested for scholars wanting to use them. In the case of European founded data collection, please provide the data set, as usual. This kind of comparative paper is more readable, cited and have a larger impact if the pre-analysis plan and Replication data are made available. 

To conclude: I am very grateful to the authors of this research. The topics addressed and the value of the dataset is undeniable. 

This is a promising paper, on a very important subject. It is not ready for publication, but it could easily be improved and published. 

The paper is well structured and my comments are finalized to improve the strength of your argument, avoid pitfalls of poor sampling in three countries, and enhance the relationship with existing literature, and the clarity of your argument. 

Author Response

This is a great paper. Full of potential. Useful. I want to see it published. But empirics is not robust. Great research question, but a weak sample. Making results not reliable. It is well written, but with poor reference to the literature. 

Thank you very much for your review, we have addressed all your comments in order to improve the quality of our manuscript.

The paper deserves to be published. It may offer a serious contribution to the existing literature. But its empirical findings need to be much stronger. Simple suggestions: cut Germany, cut Malta and cut France. You can not treat together Germany and Austria. You can not make an analysis on a so small sample of migrants in Germany, Malta and France. 

We really appreciate your suggestion and we have run all the analyses again, excluding the sampling from Germany, Malta and France as suggested. This change in our sample has implied different results, and we had also to reorient the discussions.

Please add as an appendix the full questionnaire (this is a requirement)

Thank you very much. Following your comment, we have added the full questionnaire as an appendix.

Please add as an appendix the demographics (age, gender, country of origin) of the sample (table 1 is not enough)

Thank you very much for your suggestion. The demographics of the final sample have been added as Table 1 in the main paper and demographics from the initial sample (N=1,407) have been transferred in the appendix as Supplementary Table 4

Please add in 3.1 a full presentation of the DMS scale and its questions (“eg” is not enough to present your main dependent variable), your main dependent variable.

We really appreciate your comment. We have added a full presentation of the DMS scale and its questions in lines 180-185, and the new description of the scale is:

Participants were asked the following questions: “When getting healthcare of any kind, have you ever had any of the following things happen to you because of your race or ethnicity?”, with seven adapted items: 1)You are treated with less courtesy than other people, 2)You are treated with less respect than other people 3)You receive poorer service than others, 4)A doctor or nurse acts as if he or she thinks you are not smart, 5)A doctor or nurse acts as if he or she is afraid of you, 6)A doctor or nurse acts as if he or she is better than you, 7)You feel like a doctor or nurse is not listening to what you were saying. Response categories were 1=never, 2=rarely, 3=sometimes, 4=most of the time, and 5=always. Then, all seven questions were summed and a mean score on the entire scale was computed, with higher scores indicating more perceived discrimination (range from 1-5 units).

Add the table with Cronbach diagonal results, question by question. Include only cases for countries where you have a large enough sample (so to say exclude Austria, Malta, and France)

Thank you very much. We have made the changes requested in Supplementary Table 2.

Please add in 3.1 the table with Cronbach diagonal results, question by question of the SF36 scale. Include only cases for countries where you have a large enough sample (so to say exclude Germany, Malta and France)  

Thank you very much for your comment. We have made the requested changes in Supplementary Table 3 (line 286-288).

In 3.5 it is not necessary to say you have used Stata. 

We appreciate your comment very much, and we have deleted the respective sentence.

Concerning the main results, please show your results (descriptives) on the migrant women perceived discrimination by Country and by age

Thank you very much. Following your suggestion, we added a Table (Table 3, from line 277 to 279)). It should be noted that perceived discrimination (DMS scale) is a quantitative variable and thus we cannot split migrant women by having or not having perceived discrimination.

Line 347: not useful here to refer to psychometric properties. It is confusing. 

Thank you very much for your appreciation. We tried to clarify the first sentence of our paragraph: “To our knowledge, this is the first study using the DMS questionnaire among a multi-cultural group of migrants in Europe.” (line 379-381)

Concerning undocumented migrants results are robust (but exclude France, Germany & Malta). But interpretation is poor. I would look at strong interpretation in migration studies: not to waste your time, I suggest you to check the recent contribution from Desmond King et al. pointing at explaining discrimination of undocumented migrants in health (as well as school and other public welfare services in European member states in terms of institutional differentialism (or institutional differential treatment) concerning (1) compulsory assigned residency, (2) resources (included language skills) and (3) freedom of movements (related to documented/undocumented status). These factors are keys to create a socialization process towards institutional skills. It is not necessary to follow the whole theory, but just to point on the fact that serious differential treatment of these 3 spheres consolidates heavy and stable forms of devaluation, reification and stigma (King, D. et al., 2017, Assimilation, Security, and Borders in the Member States. In King, D. and P. L. Galès, Reconfiguring States in Crisis. Oxford University Press), pp. 428–50.

Thank you very much for suggesting complementary works to support our discussions. However, since the results do not point out the relationship between health discrimination and legal status in our population, we have removed this part of the discussion and we have focused it on the relationship between age and perceived discrimination. Nevertheless, we have followed the discussion to explain cross-national differences from line 484 to 495.

“In this regard, Robertshaw and colleagues (2017) found that the immigration status and legislative policy are a challenge for the provision of healthcare by creating or reinforcing vulnerability of marginalized groups (Ahmad et al. 2020). However, the results of our study could be understood interpreted in the light of the results of Dauvrin and colleagues (2012) who also reported insufficiencies in the actual delivery of care for undocumented migrants despite the variations in healthcare entitlement related to the immigration status across Europe. Therefore, our results also outline a complex interplay of different factors that might be worsening the provision of healthcare for immigrant patients beyond their legal status in the host country”

Line 395 I will not use the concept of readiness to accommodate. It seems tautological to me. Why health system they do not accommodate undocumented migrants? Because they are not ready to. Poor explanation. There are three main ways to look at health system capacity to cope with undocumented migrants in public health literature: the first is related to the organizational dimension – it has proportionally received the largest amount of attention by the literature explaining the strategic decision-making behaviour among NGOs, Unions, Churches and pro-migrant interest groups that seek to influence system openness and response to undocumented migrants health demand. The second set of factors relates to the path-dependent universalism of the health system and to the rules and boundary mechanisms that have contributed to the construction of its reception rules via opposition/distinction of deserving “clients”, see Vanhala (2009). The third set of factors has been explored by authors looking for resources for adaptation and inclusionary improvement within this system (Jacquot in the Journal of European Public Policy 2014 has discussed this with reference to your same main discriminated groups: women and minorities) 

Thank you very much for suggesting complementary works to support our discussions. However, since the results do not point out the relationship between health discrimination and legal status in our population, we have removed this part of the discussion and we have focused it on the relationship between age and perceived discrimination. Nevertheless, we have followed the discussion to explain cross-national differences from line 484 to 495.

“In this regard, Robertshaw and colleagues (2017) found that the immigration status and legislative policy are a challenge for the provision of healthcare by creating or reinforcing vulnerability of marginalized groups (Ahmad et al. 2020). However, the results of our study could be understood interpreted in the light of the results of Dauvrin and colleagues (2012) who also reported insufficiencies in the actual delivery of care for undocumented migrants despite the variations in healthcare entitlement related to the immigration status across Europe. Therefore, our results also outline a complex interplay of different factors that might be worsening the provision of healthcare for immigrant patients beyond their legal status in the host country”

line 418: attention not to blame the victims

Thank you very much for your suggestion. We have modified the sentence and now reads like this:

“Moreover, more frequent health visits could increase the likelihood to be exposed to suffer experiences of healthcare discrimination towards them.”

line 456: ‘convenience sampling’ is not clear, it requires to be defined.

Thank you very much for your comment. We have rephrased this part (lines 478-486) as follows: The use of interpreters may have introduced additional information bias, and cultural barriers in female representation in the survey for some countries (such as Afghanistan) may have biased their responses in giving socially desirable answers, leading to underestimation of healthcare discrimination. Finally, as this is a cross-sectional study which relies on a non-random sample, causal relationships cannot be established.

line 457 the relation between sampling and socially desirable answers is not clear; usually, the link is with the interview method, not with the sampling technique

Again, thank you very much for highlight this. As we have mentioned previously, we have addressed this comment by rephrasing the paragraph on the sampling (lines 478-486).

Please clearly state if your data are publicly available, and where they can be found and requested for scholars wanting to use them. In the case of European founded data collection, please provide the data set, as usual. This kind of comparative paper is more readable, cited and have a larger impact if the pre-analysis plan and Replication data are made available. 

We direct the scholars to the project’s website for additional information. More detailed data can be made available upon request and following permission from the project’s consortium.

To conclude: I am very grateful to the authors of this research. The topics addressed and the value of the dataset is undeniable. 

This is a promising paper, on a very important subject. It is not ready for publication, but it could easily be improved and published. 

The paper is well structured and my comments are finalized to improve the strength of your argument, avoid pitfalls of poor sampling in three countries, and enhance the relationship with existing literature, and the clarity of your argument. 

Thank you very much for all your comments and for allowing us to improve the quality of our manuscript.

Round 2

Reviewer 3 Report

Thank you very much to the author(s) for the great work done. I'm very satisfied with the new results. Having cut countries with a small sample made your results much more robust. I've really appreciated the modifications, and your detailed answers to my suggestion. Great job! The paper is almost ready to be published and will be an important contribution. But it still requires a bit of discussion and interpretation of the results. Your point is that there are "insufficiencies in the actual delivery of care for undocumented migrants". But this is too generic to interpret your results. Is it a problem of quantity of care delivery? Where is it coming from? This is why I insist that you can sipmiy look at at contemporary robust theory in migration research to avoid mere description. I insist on the relevance of Desmond King et al. contribution for Oxford University Press (chapter on integration, assimilation and the refugee reception in EU member states) to discuss the relationship between healthcare discrimination and collective services in terms of institutional differential treatment concerning (1) compulsory assigned residency, (2) resources (included language skills) and (3) freedom of movements (related to documented/undocumented status), consolidating heavy and stable forms of devaluation, reification and stigma with consequences on health (King, D. et al., 2017, Assimilation, Security, and Borders in the Member States. In King, D. and P. L. Galès, Reconfiguring States in Crisis. Oxford University Press), pp. 428–50.

I really hope you will take this point seriously, and your paper will be a very important contribution in the field

Author Response

Dear Reviewer 3, 

First of all, thank you very much for your words regarding the potential of our manuscript. We have extended the discussion explaining our point "insufficiencies in the actual delivery of care for undocumented migrants", and now the paragraph reads like this (line 445 - 457):

"However, the results of our study could be interpreted in the light of the results of Dauvrin and colleagues (2012) who also reported insufficiencies in the actual delivery of care for undocumented migrants despite the variations in healthcare entitlement related to the immigration status across Europe, suggesting that even in countries with “minimum rights”, health professionals may consider treating undocumented migrants more important than abiding by law (“pragmatic health professional”). For that reason, our results might be outlining a complex interplay of different factors that might be worsening the provision of healthcare for migrant patients in the host country beyond their legal status, as this is not a general result and differences appear between countries in our study."  

Moreover, we have gone through the chapter that you highly recommended and we have further developed our discussions considering your suggestion (line 458 - 468):

"Although specific migrant groups have reported experiencing discrimination in healthcare, ethnic discrimination, and translation services are still under-researched topics in Europe. In this regard, our results could be also interpreted from a structural and organizational point of view for healthcare delivery. Indeed, King and colleagues (2017) argue how (1) compulsory as-signed residency, (2) resources (included language skills), and (3) freedom of movements (related to documented/undocumented status) could be consolidating heavy and stable forms of devaluation, reification, and stigma, denying the access to healthcare for certain groups with negative consequences on the health of migrants." 
